**Data Availability Statement:** Data cannot be shared publicly because we do not have permission from the Research Ethics Committee to

# Health risk behaviours among people with severe mental ill health during the COVID-19 pandemic: Analysis of linked cohort data

Emily Peckham[1]*, Victoria Allgar[2], Suzanne Crosland[1], Paul Heron[1], Gordon Johnston[3], Elizabeth Newbronner[1], Panagiotis Spanakis[1], Ruth Wadman[1], Lauren Walker[1], Simon Gilbody[1,2]

1 Mental Health and Addiction Research Group, University of York, Heslington, United Kingdom, 2 Hull York Medical School, Heslington, United Kingdom, 3 Independent Peer Researcher, United Kingdom

* emily.peckham@york.ac.uk

## Abstract

### Background

People with severe mental ill health (SMI) experience a mortality gap of 15–20 years. COVID-19 has amplified population health inequalities, and there is concern that people with SMI will be disproportionately affected. Understanding how health risk behaviours have changed during the pandemic is important when developing strategies to mitigate future increases in health inequalities.

### Methods

We sampled from an existing cohort of people with SMI. Researchers contacted participants by phone or post to invite them to take part in a survey about how the pandemic had affected them. We asked people about their health risk behaviours and how these had changed during the pandemic. We created an index of changed behaviours, comprising dietary factors, smoking, lack of exercise, and drinking patterns. By creating data linkages, we compared their responses during pandemic restrictions to responses they gave prior to the pandemic.

### Outcomes

367 people provided health risk data. The mean age of the participants was 50.5 (range = 20 to 86, SD ± 15.69) with 51.0% male and 77.4% white British. 47.5% of participants reported taking less physical activity during the pandemic and of those who smoke 54.5% reported smoking more heavily. Self-reported deterioration in physical health was significantly associated with an increase in health risk behaviours (adjusted OR for physical health 1.59, 95%CI 1.22–2.07; adjusted OR for Age 0.99, 95%CI 0.98–1.00).

publicly share participant level data. The data that support the findings of this study are available on request from the Closing the Gap Network email: ctg-network@york.ac.uk for researchers who meet the criteria for access to confidential data.

**Funding:** EP, PS, PH, GJ, EN and SG Medical Research Council (grant reference MR/V028529). https://mrc.ukri.org/funding/ SG Economic and Social Research Council (ES/S004459/1). https://www.ukri.org SG, RW, EN and PS National Institute for Health Research (NIHR200166) https://www.nihr.ac.uk.

**Competing interests:** The authors have declared that no competing interests exist.

## Interpretation

COVID-19 is likely to amplify health inequalities for people with SMI. Health services should target health risk behaviours for people with SMI to mitigate the immediate and long lasting impacts of the COVID-19 pandemic.

## Introduction

The COVID-19 pandemic is likely to impact on public health in ways that extend beyond the number of deaths attributable to coronavirus [1, 2]. One possible impact on public health is a change in health related behaviours, which may be affected by the social measures introduced to limit the spread of the virus (pandemic restrictions) such as minimizing social contacts and staying at home. The pandemic restrictions, coupled with daily news coverage of the number of COVID-19 related deaths, are likely to impact on people in a variety of ways. For example, the perceived risks of contracting COVID-19 may provide a 'teachable moment' that acts as a prompt to encourage people to make changes to their behaviour; such as cutting down on drinking alcohol and smoking or increasing physical activity [3]. On the other hand, concerns about COVID-19 may lead people to avoid undertaking physical activity outdoors and lead more sedentary lives. Additionally concerns around contracting the virus, increased social isolation or the stress as a result of the pandemic restrictions may lead some people to drink and smoke more and find it more difficult to make healthy food choices [4]. Furthermore, access to healthy food may have become more difficult due to fears about supermarkets, difficulty in assessing online food shopping and closure of cafes and community projects where people may have accessed healthy meals. In addition, the closure of gyms and swimming pools may have impacted on physical activity levels [5].

Understanding the impacts of the pandemic restrictions on health-related behaviours is important for assessing the wider public health consequences of the pandemic[6]. In turn this is also helpful in informing how these impacts can be mitigated. Addressing health risk behaviours remains a public health priority [7, 8]. Smoking, excessive alcohol consumption, physical inactivity and obesity [9–11] are some of the leading causes of disease and premature death worldwide. The effects of the pandemic on health related behaviours in the general population have been mixed. It has been reported that in the early stages of the pandemic 300,000 quit (smoking) for Covid [12], yet the percentage of people who smoke in the general population remains very similar to the figure before the pandemic [13]. A similar picture has been reported for alcohol consumption, with high risk drinkers being more likely to report trying to reduce their alcohol consumption at the start of the pandemic. However, the prevalence of high risk drinking has increased [13]. A study looking at behaviours around weight and physical activity has shown comparable results, with some people reporting having undertaken more physical activity and others reporting taking less [4]. A study in Spain has shown that, the early stages of the pandemic, people who meet the WHO guidelines for physical activity are have lower perceived anxiety and lower perceived worse mood [14]. Importantly, studies have generally demonstrated that the pandemic restrictions may have a disproportionately large influence on behaviours which cause weight gain among adults with a pre-existing higher BMI [4], and among people from lower socio-economic groups and with pre-existing conditions [15].

It is possible that the pandemic and its associated restrictions will amplify already existing health inequalities; people with severe mental ill health (SMI) are one such group that may be adversely affected [16]. People with SMI currently experience a mortality gap of 15–20 years when compared to people without SMI [17, 18]. One of the major drivers of this health gap is

preventable physical health conditions linked to behavioural risk factors such as smoking, low physical activity and nutritionally-poor diet. Although studies have explored the impacts of the pandemic and pandemic restrictions on the general population [4, 13, 15, 19] there have, to the best of our knowledge, been no studies that have explored the health related behaviours among people with SMI.

People with SMI are an especially vulnerable group who already experience significant health inequalities. Understanding the specific difficulties encountered by people with SMI becomes critical in identifying how to best to mitigate the risks of them experiencing further deprivation and disadvantage during the current pandemic and beyond. People with SMI are also under-researched with respect to health and health related behaviours, and we were able examine this issue with respect to the COVID pandemic using a large clinical cohort which was recruited in the years immediately prior to the pandemic restrictions [20].

Therefore, the aim of this study was to examine self-reported changes in a range of health related behaviors before vs during the pandemic in a transdiagnostic sample of people with SMI. This is an exploratory study rather than being derived from a theoretical framework.

## Design

The Closing the Gap (CtG) study is a large (n = 9, 914) transdiagnostic clinical cohort recruited between April 2016 and March 2020. Participants have documented diagnoses of schizophrenia or delusional/psychotic illness (ICD 10 F20.X & F22.X or DSM equivalent) or bipolar disorder (ICD F31.X or DSM-equivalent). The composition of the CtG cohort has previously been described [20], and the data at inception included descriptions of self-reported alcohol consumption, consumption of fruit and vegetables, participation in physical activity, smoking behavior and e-cigarette use.

We were funded to explore the impact of the COVID-19 pandemic in a sub-section of the CtG clinical cohort and we identified participants for Optimising Well-being in Self-Isolation study (OWLS) [21]. To ensure that the OWLS COVID-19 sub-cohort was captured a range of demographics we created a sampling framework based on gender, age, ethnicity and whether they were recruited via primary or secondary care. OWLS participants were recruited from 17 mental health trusts (across urban and rural settings in England).

## Recruitment and participants

Ethical approval was granted by the Health Research Authority North West—Liverpool Central Research Ethics Committee (REC reference 20/NW/0276). To be eligible to take part in OWLS COVID-19 study, people had to be aged 18 or over, to have taken part in CtG study and have consented to be contacted again to be invited to further research. This enabled us to create longitudinal data linkage and to rapidly identify participants during the COVID-19 pandemic.

## Procedure

People who met the eligibility criteria were contacted by telephone or letter and invited to take part in the OWLS COVID-19 study. Those who agreed to take part were provided with a range of options; i. to carry out the survey over the phone with a researcher, ii. to be sent a link to complete the survey online or iii. to be sent a hard copy of the questionnaire in the post to complete and return. The questionnaire can be found in S1 File. In the questionnaire before the pandemic refers to before the pandemic restriction came into force in the UK on the 23rd March.

## Changes in health related behaviours

Participants were asked about five health related behaviours; (1) alcohol consumption, (2) levels of physical activity, (3) fruit and vegetable consumption, (4) vaping and e-cigarette use, and (5) smoking. For each health related behaviour participants were asked about how often they engaged in that behaviour compared to before the pandemic and could chose from the following options; I don't do that in general, more than before, about the same or less than before. For physical activity and fruit and vegetable consumption a response of 'less than before' was considered to be a negative change in behaviour and for drinking, smoking a response of more than before was considered to be a negative change in behaviour. A cumulative index of harmful health related behaviours were calculated from 0 to 4, [22] we did not include vaping in this metric as vaping could be seen as a positive change if it led to a harm reduction or abstinence from more harmful tobacco use [23].

## Self-reported changes in health during pandemic restrictions

For both physical health and mental health study participants were asked 'compared to your life before the pandemic restrictions, how would you rate your health in general, ' with the following response options; 'better than before', 'about the same', 'worse than before', 'not sure/ don't know'. After excluding the participants who reported 'not sure/ don't know', a binary variable was derived coded as 'decline in health' (including those reporting worse than before) and 'no decline' (including those reporting better than before or about the same).

## Wellbeing

To align with general population estimates of health related behaviour, study participants were asked four questions taken from the ONS Health and Lifestyle Survey (HLS) [24] about how they had been feeling with a response option scored on a Likert scale of 0–10. The questions asked were; (1) overall how satisfied are you with your life, (2) overall to what extent do you feel that the things you do in your life are worthwhile, (3) overall how happy did you feel yesterday and overall how anxious did you feel yesterday. For the first three questions a score of 0 indicated not at all and a score of 10 completely and for the final question a score of 0 indicated not at all anxious and 10 indicated completely anxious. After reversing the scores for the anxiety question the scores for the four questions were totaled to give a total wellbeing score (0–40), with higher scores indicating greater wellbeing.

## Ethnicity

Ethnicity was used to define a binary minority variable (BAME (Black, Asian and minority ethnic) and non-BAME). As people form BAME backgrounds are more at risk from COVID-19 [25] we wanted to determine whether they were also more at risk from health risk behaviours.

## Indices of multiple deprivation

We used postcodes already collected at the point of inception to the CtG study to determine levels of deprivation using the level of deprivation assigned to postcodes by the Ministry of Housing, Communities and Local Government [26]. Scores are given between 1 and 10 and then reduced these scores to give five possible outcomes; very high deprivation (1 and 2), high deprivation (3 and 4), moderate deprivation (5 and 6), low deprivation (7 and 8) and very low deprivation (9 and 10).

## Statistical analysis

The study analysis plan was registered on Open Science Framework (available at https://doi.org/10.17605/OSF.IO/E3KDM—section 2.1). Analyses were undertaken using SPSS v.26. Descriptive statistics were used to describe sociodemographic characteristics, wellbeing, changes in physical and mental health, health related behaviours, and whether the person was currently receiving support from mental health services. We planned to test (using multiple regression) whether the cumulative index of harmful behaviours was associated with wellbeing and deterioration in physical or mental health after adjusting for age, gender, minority status, socio-economic deprivation and being seen in secondary care. However, as the outcome variable was a count variable (0–4) with 40.3% scoring 0 we used a Poisson regression instead. Predictors were examined individually in univariate models (unadjusted models) before entered into the multivariate model (adjusted model).

## Results

Between July and December 2020 367 people from the CtG study were recruited to the OWLS COVID-19 study. From the 2,932 participants in the CtG study that were eligible to be invited to OWLS, we selected a sample of 1,166 (39.8%) participants to attempt to contact and successfully contacted 688 (59%). The survey was completed by 367 participants (31.5% of those eligible to be invited and 53.3% of those successfully contacted) see Fig 1—Flow Diagram.

Table 1 describes the socio-demographic characteristics of OWLS participants. The mean age was 50.5 (range = 20 to 86, SD ± 15.69) with 51.0% male and 77.4% white British.

Table 2 shows the self-reported health related behaviours measured in the CTG study prior to the pandemic. It also shows the changes in health related behaviours and mental and physical health measured in the OWLS COVID-19 study since the beginning of the pandemic. In terms of negative behaviour changes, 29.1% of those who drink reported drinking more than before the pandemic, 54.5% of those who smoke reported smoking more, 47.5% who take part in physical activity reported doing less physical activity and 21.2% of those who eat fruit and vegetables reported eating less fruits and vegetables.

Table 3 shows the cumulative index of harmful health related behaviour changes reported by participants. 40.3% (148/367) reported no negative behaviour changes, while out of the 56.4% (207/367) who did, 62.8% reported 1 negative behavior change.

We found that self-reported deterioration in physical health was significantly associated with an increase in behaviours known to be harmful to health (Table 4), (adjusted OR for physical health 1.59, 95%CI 1.22–2.07).

## Discussion

In the transdiagnostic study sample of adults with severe mental ill health (N = 367) we explored behaviours known to be harmful to health and how these may have changed during the COVID-19 pandemic. To the best of our knowledge this is the first study to explore health related behaviours in this population. Prior to the pandemic 20.2% of participants reported that they did not take part in physical activity, this is compared to 18% who reported that they did not take part in physical activity during the pandemic. Of those that were physically active during the pandemic 47.5% report that they are doing less than they did prior to the pandemic. This compares to 36.2% of adults in the general population who reported taking part in less physical activity during the pandemic [15]. This is a cause for concern in a population with already low baseline levels of physical activity and high levels of sedentary behaviour [27]. Furthermore, prior to the pandemic only 4.4% of CtG study participants were meeting government guidelines of five portions of fruit and vegetables per day. During the pandemic 20.4% of

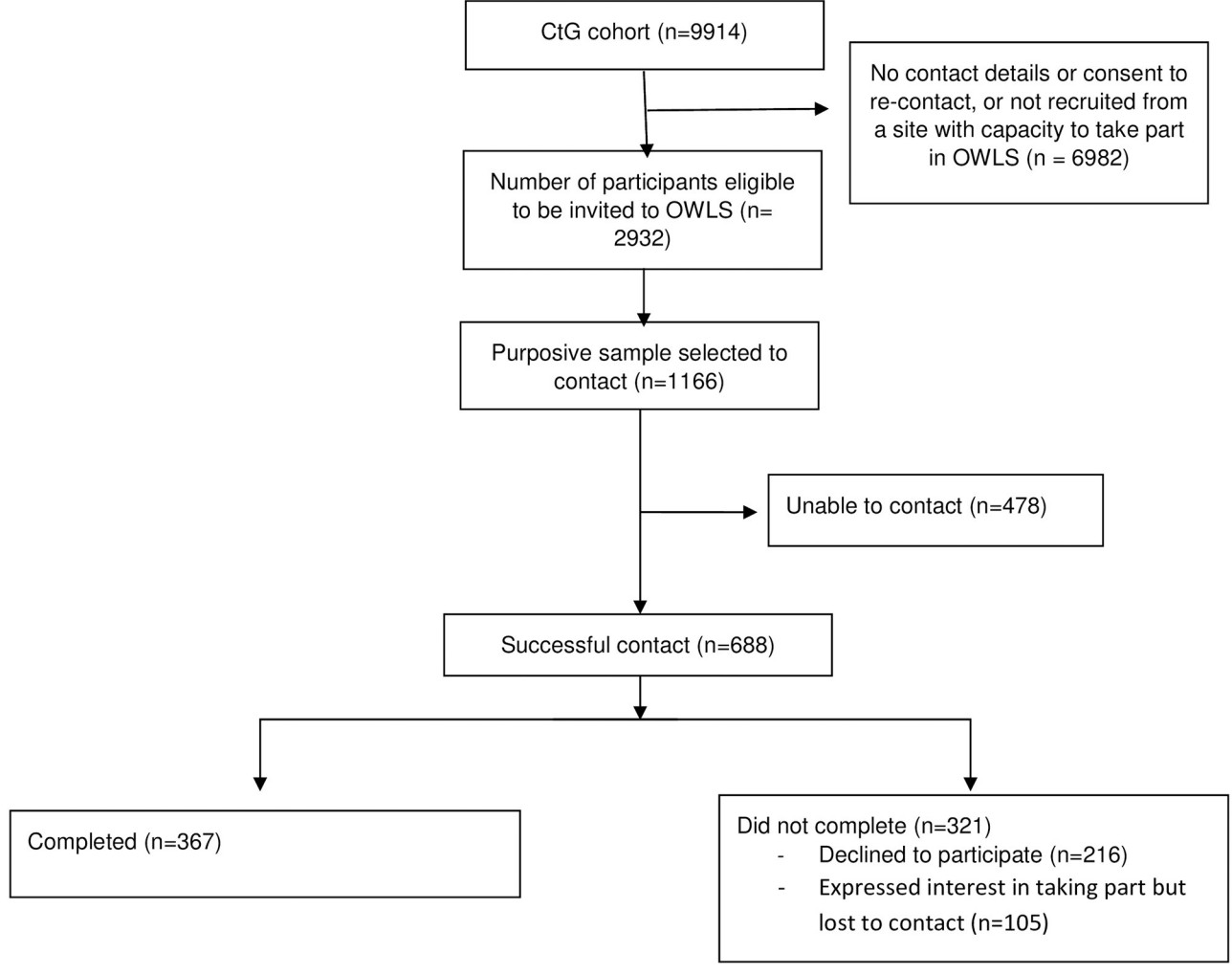

**Fig 1. Flow diagram.**

people reported eating fewer than before the pandemic. This is a slight improvement over a general population sample (4) that reported 35% of respondents saying they were eating a healthy balanced diet less than before the pandemic. However, it should be borne in mind the low baseline that people with SMI are starting from in terms of fruit and vegetable consumption.

In terms of smoking, the number of people who reported being smokers during the pandemic was nearly double that of the general population. Twenty seven percent in our sample compared to 14.8% in a general population representative sample [19]. However this number has decreased slightly from when participants in the OWLS COVID-19 study were recruited to the CtG cohort (between 2016 and 2020), when the percentage that smoked was 31.4%. Of concern is the fact that of those who do smoke 54.5% were smoking more during the pandemic. More encouragingly, we found that drinking alcohol is of relatively low prevalence amongst our study sample, prior to the pandemic 40.1% of people reported not drinking alcohol compared to 53.7% who reported not drinking alcohol during the pandemic. However, of those that do drink 29.1% were drinking more during the pandemic. These results however should be interpreted with caution as some people in this study are taking medication that

**Table 1. Demographic characteristics of OWLS COVID-19 study SMI population during pandemic restrictions.**

| Characteristic | N(%) total n = 367 |
|---|---|
| **Self- reported gender** | |
| Male | 187 (51.0) |
| Female | 174 (47.4) |
| Transgender | 6 (1.6) |
| Missing | 0 (0.0) |
| **Age** (mean, range, SD) | 50.5 (20–86) (15.7) |
| **Ethnicity** | |
| White British | 284 (77.4) |
| Other white | 18 (4.9) |
| Mixed white/ black | 5 (1.4) |
| Mixed white/ Asian | 5 (1.4) |
| Other mixed | 4 (1.1) |
| Asian | 24 (6.5) |
| Black | 15 (4.1) |
| Other non-white | 12 (3.3) |
| Missing | 0 (0.0) |
| **Index of deprivation** | |
| Very high deprivation | 97(26.4) |
| High deprivation | 81 (22.1) |
| Moderate deprivation | 67 (18.3) |
| Low deprivation | 55 (15.0) |
| Very low deprivation | 52 (14.2) |
| Missing | 0 (0.0) |
| **Being seen in secondary** | |
| **care** | |
| Yes | 224 (61.0) |
| No | 139 (37.9) |
| Missing | 0 (0.0) |
| **Wellbeing score (mean, SD)** | 23.1 (8.74) |

contradicts the consumption of alcohol and therefore it may be that some people who drink alcohol did not want to report that they consumed alcohol.

Over half of the OWLS COVID-19 study sample reported greater levels of behaviours known to be harmful to health during the pandemic restrictions. In particular, they reported smoking more, drinking more, eating less fruit and vegetables or taking part in less physical activity. The majority of people had only one increase in behaviour harmful to health, although just under a third had an increase in two behaviours known to be harmful to health. When we explored associations with behaviours known to be harmful to health we found that a reporting a deterioration in physical health was associated with increased health risk behaviour. However, reporting a deterioration in mental health was not found to be associated with behaviours known to be harmful to health. Socio-demographic factors (BAME, IMDD, gender) were also not found to be associated with behaviours known to be harmful to health. We do not know whether an increase in health risk behaviours led to a deterioration in physical health or whether a deterioration in physical health led to people being less able to take care of themselves, and this relationship is likely to be bidirectional. The relationship between physical health and health related behaviours highlights

**Table 2. Self-reported health and health risk behaviours for the OWLS COVID-19 study participants.**

|  | N (%) total n = 367 |
|---|---|
| **Drinking and alcohol use** | |
| **Pre-COVID-19** | |
| Doesn't drink alcohol | 147 (40.1) |
| Drinks more than 14 units per week | 28 (7.6) |
| Missing | 64 (17.4) |
| **During COVID-19** | |
| **I don't do that in general** | 197 (53.7) |
| **Those who report drinking alcohol** | 165 (45.0) |
| More than usual | 48 (29.1) |
| About the same | 72 (43.6) |
| Less than usual | 45 (27.3) |
| Missing | 5 (1.4) |
| **Smoking and tobacco use** | |
| **Pre-COVID-19** | |
| People who smoke | 114 (31.1) |
| People who used to smoke | 110 (30.0) |
| People who have never smoked | 140 (38.1) |
| Missing | 3 (0.8) |
| **During COVID-19** | |
| **I don't do that in general** | 259 (70.6) |
| **Those who report smoking** | 99 (27.0) |
| More than usual | 54 (54.5) |
| About the same | 33 (33.3) |
| Less than usual | 12 (12.1) |
| Missing | 9 (2.5) |
| **Vaping and e-cigarette use** | |
| **I don't do that in general** | 309 (84.2) |
| **Those who report vaping** | 47 (12.8) |
| More than usual | 21 (44.7) |
| About the same | 20 (42.6) |
| Less than usual | 6 (12.8) |
| Missing | 11 (3.0) |
| **Taking part in physical activity** | |
| **Pre-COVID-19** | |
| Never take part in physical activity | 74 (20.2) |
| Takes part in physical activity | 288 (78.5) |
| Met government guidelines | 156 (42.5) |
| Missing | 5 (1.4) |
| **During COVID-19** | |
| **I don't do that in general** | 67 (18.3) |
| **Those who report taking part in physical activity** | 295 (80.4) |
| More than usual | 61 (20.7) |
| About the same | 94 (31.9) |
| Less than usual | 140 (47.5) |
| Missing | 5 (1.4) |
| **Eating 5 portions of fruit and veg per day** | |
| **Pre-COVID-19** | |

(*Continued*)

**Table 2.** (Continued)

|  | N (%) total n = 367 |
|---|---|
| Don't eat fruit and veg | 63 (17.2) |
| Eats fruit and veg | 296 (80.7) |
| Met government guidelines | 16 (4.4) |
| Missing | 8 (2.2) |
| **During COVID-19** |  |
| **I don't do that in general** | 77 (21.0) |
| **Those who report eating fruit and veg** | 274 (77.4) |
| More than usual | 38 (13.4) |
| About the same | 188 (66.2) |
| Less than usual | 58 (20.4) |
| Missing | 6 (1.6) |
| **Self reported global mental health** |  |
| No deterioration | 210 (57.2) |
| Deterioration | 148 (40.3) |
| Missing | 2 (0.5) |
| **Self-reported global physical health** |  |
| No deterioration | 236 (64.3) |
| Deterioration | 118 (32.2) |
| Missing | 2 (0.5) |

the importance of developing and maintaining strategies to improve the physical health of people with SMI. In addition, younger age was found to be a potential predictor of new behaviours known to be harmful to health under the pandemic. This may be explained by a number of factors, it could be that older people perceive themselves to be more at risk from COVID-19 and are therefore less likely to adopt additional behaviours known to be harmful to health. Younger people may have been more likely to take part in physical activity, which had to stop as the gyms closed and sports clubs couldn't meet. Another possibility is that people with SMI who engage in negative health behaviours are more likely to have a shorter life expectancy, and not survive into older age. This could be a manifestation of the healthy survivor phenomenon observed in cohort studies [28].

This OWLS COVID-19 study has several strengths. First, we report on a group of people who are underrepresented in general population surveys, and yet have some of the highest levels of unmet health needs [29]. People were directly approached to take part in this study using a sampling framework that allowed people from a range of demographics to be invited rather than a self-selected group, which is a risk for online surveys and can lead to selection

**Table 3. Cumulative index of self-reported behaviours harmful to health for the OWLS COVID-19 study participants.**

|  | n (%) total n = 367 |
|---|---|
| **No reported changes in behaviours known to be harmful to health** | 148 (40.3) |
| **One or more changes in behaviours known to be harmful to health** | 207 (56.4) |
| 1 | 130 (62.8) |
| 2 | 67 (32.4) |
| 3 | 9 (4.3) |
| 4 | 1 (0.5) |

**Table 4. Associations between socio-demographic factors, health related variables and behaviours known to be harmful to health.**

|  | Odds ratio (95% CI) | P |
|---|---|---|
| Age | 0.99 (0.98–1.00) | 0.03 |
| Wellbeing score | 1.00 (0.98–1.01) | 0.72 |
| **Self reported gender** | | |
| Male | 1 | |
| Female | 1.01 (0.79–1.31) | 0.89 |
| Transgender | 0.67 (0.25–1.85) | 0.44 |
| **Self-reported ethnicity** | | |
| Non BAME | 1.06 (0.75–1.50) | 0.74 |
| BAME | 1 | |
| **Level of deprivation** | | |
| Very low deprivation | 1.13 (0.78–1.63) | 0.51 |
| Low deprivation | 0.84 (0.57–1.24) | 0.38 |
| Moderate deprivation | 0.79 (0.54–1.15) | 0.22 |
| High deprivation | 0.91 (0.65–1.28) | 0.59 |
| Very high deprivation | 1 | |
| **Ongoing contact with secondary care** | | |
| Yes | 1.04 (0.80–1.34) | 0.78 |
| No | 1 | |
| **Self-reported decline in physical health** | | |
| Decline | 1.59 (1.22–2.07) | 0.00* |
| No decline | 1 | |
| **Decline in mental health** | | |
| Decline | 1.28 (0.96–1.69) | 0.09 |
| No decline | 1 | |

*indicates statically significant difference.

bias. The problem of potentially unrepresentative self-selected samples in COVID-19 studies has been shown to be a particular limitation in mental health research [30]. Second, the OWLS COVID-19 study participants had previously been recruited to a large transdiagnostic study of people with SMI, and had consented to be contacted about future research. This allowed us to recruit to the OWLS COVID -19 study rapidly in response the COVID-19 pandemic, whilst still using a sampling strategy. This study does however have some weaknesses. First, whilst every effort was made to include the most vulnerable people from what is a vulnerable group we were rarely able to recruit from inpatient settings or people under the care of assertive outreach teams. When potential participants were on an inpatient ward we asked their care team if they could support the person to take part where it was appropriate to do so. Unfortunately there were very few cases where this was possible. We also attempted to include people in community supported living facilities and asked their care team to support them to take part, in some cases this was possible but in others it proved more difficult. Secondly, the survey relies on self-reported data and people may not accurately recall or may not want to accurately answer the question/s. However this is true for any form of survey that does not have a way of verifying the data provided. Thus the results should be interpreted with this in mind. Thirdly, the design used in this study does not allow us to determine the causes of any changes in health related behaviours so we cannot be certain as to why behaviours might have changed.

## Conclusion

People with SMI are considered clinically vulnerable, and these concerns predate the COVID-19 pandemic. The results of this study highlight the additional threats to the health of a group of people who already experience profound health inequalities. SMI is listed as an underlying health condition which puts them at greater risk during the COVID-19 pandemic and many people with SMI also have co-morbid physical health conditions which are additional risk factors also listed as underlying health conditions, for example diabetes and obesity [31]. The results around smoking and exercise are especially concerning, as being a smoker and low levels of physical activity are risk factors for poor COVID-19 outcomes [32, 33] and smoking is a possible risk factor for greater transmission [34].

This evidence presented in this paper further emphasises the need for services that target health risk behaviours, such as smoking, diet and physical exercise, for people with SMI to mitigate these high risks of adverse outcomes in the presence of COVID-19 and in the longer term.

## Supporting information

**S1 Checklist.**
(DOCX)

**S1 File.**
(PDF)

## Acknowledgments

We would like to thank the participants in the OWLS study and NHS mental health staff for their support with this study. Particular thanks go to our partner trust Tees, Esk and Wear Valleys NHS Foundation Trust.

## Author Contributions

**Conceptualization:** Emily Peckham, Suzanne Crosland, Paul Heron, Gordon Johnston, Elizabeth Newbronner, Panagiotis Spanakis, Ruth Wadman, Lauren Walker, Simon Gilbody.

**Data curation:** Panagiotis Spanakis.

**Formal analysis:** Emily Peckham, Victoria Allgar, Panagiotis Spanakis.

**Funding acquisition:** Emily Peckham, Paul Heron, Gordon Johnston, Elizabeth Newbronner, Panagiotis Spanakis, Simon Gilbody.

**Investigation:** Emily Peckham, Suzanne Crosland, Paul Heron, Gordon Johnston, Panagiotis Spanakis, Ruth Wadman, Lauren Walker.

**Methodology:** Emily Peckham, Victoria Allgar, Paul Heron, Elizabeth Newbronner, Panagiotis Spanakis, Ruth Wadman, Lauren Walker, Simon Gilbody.

**Supervision:** Victoria Allgar.

**Writing – original draft:** Emily Peckham, Lauren Walker, Simon Gilbody.

**Writing – review & editing:** Victoria Allgar, Suzanne Crosland, Paul Heron, Gordon Johnston, Elizabeth Newbronner, Panagiotis Spanakis, Ruth Wadman.

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
