## [Decision Letter · Decision Letter 0]

15 Jul 2021

PONE-D-21-09538

Health risk behaviours among people with severe mental ill health during the COVID-19 pandemic: analysis of linked cohort data

PLOS ONE

Dear Dr. Peckham,

Thank you for submitting your manuscript to PLOS ONE. After careful consideration, we feel that it has merit but does not fully meet PLOS ONE’s publication criteria as it currently stands. Therefore, we invite you to submit a revised version of the manuscript that addresses the points raised during the review process.

We look forward to receiving your revised manuscript.

Kind regards,

Stanton A. Glantz, PhD

Academic Editor

PLOS ONE

Journal Requirements:

2.

We note that you have indicated that data from this study are available upon request. PLOS only allows data to be available upon request if there are legal or ethical restrictions on sharing data publicly. For information on unacceptable data access restrictions, please see http://journals.plos.org/plosone/s/data-availability#loc-unacceptable-data-access-restrictions.

3. We note you have included a table to which you do not refer in the text of your manuscript. Please ensure that you refer to Table 1 and 7 in your text; if accepted, production will need this reference to link the reader to the Table.

Reviewers' comments:

Reviewer's Responses to Questions

**Comments to the Author**

1. Is the manuscript technically sound, and do the data support the conclusions?

Reviewer #1: Partly

Reviewer #2: Partly

2. Has the statistical analysis been performed appropriately and rigorously? 

Reviewer #1: No

Reviewer #2: Yes

3. Have the authors made all data underlying the findings in their manuscript fully available?

Reviewer #1: No

Reviewer #2: Yes

4. Is the manuscript presented in an intelligible fashion and written in standard English?

Reviewer #1: Yes

Reviewer #2: Yes

5. Review Comments to the Author

Reviewer #1: Introduction

References supporting the claim that people with SMI are at increased risk during a pandemic are not provided; while this statement has face validity, the authors should provide literature supporting this claim, even if it is only through referencing similar situations.

Methods

The data appear to rely solely on self-report of health behaviors; it is not clear whether these reports reflect actual behavior (historically, for example, self-reports about consumption of produce poorly reflect actual consumption). Better description of the OWLS subcohort is needed; current description does not meet research checklist guidelines (see Equator Network for details). Authors do not fully describe the questions in the survey instrument: no supplement with these details was included in the manuscript provided for review, nor was one mentioned in the text. Scaling of global health and wellbeing is rudimentary and given the potential overlap in underlying concepts, some estimation indicating whether these measures are loading together would be critical (e.g. factor analysis, SEQ). There is no indication of why a binary measure of race/ethnicity would be valid in this sample. No research supporting the decision to measure relative deprivation by postcode analysis is provided. Capitalize Poisson. 40.3% is not a majority; it is (apparently) a plurality. Critically, the specific dates used to define pre- and post-COVID periods are not provided.

Results

Response rate is not provided (please use standardized response rate from AAPOR or another comparable source). Race/ethnicity was proposed as a binary variable in methods but broken out by additional categories in results; this is inconsistent, and the choice of categories is not well-explained. Table 1 is difficult to read given that counts and percentages are placed in a single column; in addition, standard deviations for measures are not provided. Table 2 has similar issues, in addition, placing pre- and post-COVID variables vertically rather than horizontally makes it difficult to for the reader to assess changes. Table 4: unclear why univariate models are included at all; table formatting is challenging to read (eg CIs spread across multiple lines, significant associations not distinguished from non-significant with asterisks/bolding, spacing spreads results across multiple pages), and dependent variable is not explicitly stated. Associations between health-harming behaviors and deterioration raised questions of causality. It is unclear to what extent these behaviors would be unique to people with SMI.

Discussion

Comparisons in discussion suggest decreases in physical activity/increase in smoking but do not test for statistical significance, making it unclear whether these are meaningful. The sources of these claims are not linked to the results section (where the tests for significance should be reported). Links should be included as references not in the text of the manuscript. Conclusions re: the need for health services are not clearly linked to results.

Data availability

PLOS 1 historically requests that data underlying analysis be made available through a public repository, or, if held by an organization that limits access to data for reasons of confidentiality, for example a government agency that retains protected health information, that a detailed description of how to submit a formal data request and obtain a standardized dataset be provided, e.g. through a website with an explanation of what forms must be completed and what variables to request. The statement that data are “available upon request from the corresponding author” does not meet either standard.

Reviewer #2: This is an interesting manuscript dealing with a timely topic. Overall, it is well-conducted, although the authors should address several points.

Abstract

-Some information on the participants features should be provided (sex, age)

-There is no significant association between younger and an increase in health-risk behaviours since the IC comprises 1. This is not worth mentioning here.

Introduction

-It is mainly well-conducted, but I think that you should mention more relevant studies to support your argument. For instance, since the pandemic has affected health-risk behaviours in general population, this might be exacerbated in people with mental illness. Also, as a consequence of a lack of physical activity, mental health might be deteriorated

https://pubmed.ncbi.nlm.nih.gov/32581985/

https://pubmed.ncbi.nlm.nih.gov/32793013/

-An initial hypothesis based on your rationale should be provided

Methods

-Could you provide more information on your sampling framework. It seems as if you have been selecting individuals by those mentioned criteria, but that is not sampling.

-According to the procedure, only those who accepted were included in the study, thus there is a participation rate, I guess.

-Were the questions to assess health-related behaviours or self-reported changes validated?

-The way you explain why you use Poisson regression is too wordy. You don´t need to explain your prior plans

-Why did you select such covariates?

-Did you have missing values?

Results

-When you report decimals in your Tables, you should stick to always the same number of those

-Table 4 needs to report the confounders you adjusted your models

Discussion

-Your study has three clear bias that should be mentioned: recall bias, selection bias (not real sampling), and causal bias, since your design does not allow to reach causes

Conclusion

-It is the first time I see citations in a Conclusion

6. PLOS authors have the option to publish the peer review history of their article (what does this mean?). If published, this will include your full peer review and any attached files.

Reviewer #1: No

Reviewer #2: No

---

## [Author Response · Author response to Decision Letter 0]

23 Aug 2021

We are not able to share a de-identified data set as we do not have consent from the research participants to do this. We have checked with the GDPR team at the University of York and they have advised us that we cannot upload this data to a public repository without explicit consent from the study participants. They did however advise us that we could upload an aggregated data set which we are planning to do. We have consulted with the Research Ethics Committee that approved our study and they confirmed that it would be acceptable for us to upload the aggregated dataset to a public repository. We are currently preparing the data set for uploading and will send you the link as soon as it is done. As this is COVID-19 research, we didn’t want to delay the processing of the responses to the other comments hence we have submitted them now. 

Data requests for the full dataset may be sent to the Closing the Gap Network email: ctg-network@york.ac.uk whose Steering Committee manage our data access requests. 

Reviewer One 

Introduction

1. References supporting the claim that people with SMI are at increased risk during a pandemic are not provided; while this statement has face validity, the authors should provide literature supporting this claim, even if it is only through referencing similar situations.

We have added a citation to support the statement that people with SMI may be at increased risk. (https://www.ncbi.nlm.nih.gov/pmc/articles/PMC7250778/)

Methods 

2. The data appear to rely solely on self-report of health behaviors; it is not clear whether these reports reflect actual behavior (historically, for example, self-reports about consumption of produce poorly reflect actual consumption).

We agree that self-reported data is not always accurate however there is no clear way of collecting objective data on health risk behaviours, e.g. smoking status can be verified by CO measurement but this cannot tell us how many cigarettes a person is smoking per day. This means that we have had to reply on self-report data. We have stated this as a limitation in our discussion.

“Secondly, the survey relies on self-reported data and people may not accurately recall or may not want to accurately answer the question/s. However this is true for any form of survey that does not have a way of verifying the data provided. Thus the results should be interpreted with this in mind.”

3. Better description of the OWLS subcohort is needed; current description does not meet research checklist guidelines (see Equator Network for details). 

We apologise for this oversight. We have now attached a completed STROBE checklist, added additional text (page 21) to explain how the sub-cohort relates to the original cohort and added a flow diagram (Figure 1). 

“From the 2,932 participants in the CtG study that were eligible to be invited to OWLS, we selected a sample of 1,166 (39.8 %) participants to attempt to contact and successfully contacted 688 (59%). The survey was completed by 367 participants (31.5% of those eligible to be invited and 53.3% of those successfully contacted) see Figure 1.”

4. Authors do not fully describe the questions in the survey instrument: no supplement with these details was included in the manuscript provided for review, nor was one mentioned in the text. 

We are sorry for this omission. We have included the survey as a supplementary file

5. Scaling of global health and wellbeing is rudimentary and given the potential overlap in underlying concepts, some estimation indicating whether these measures are loading together would be critical (e.g. factor analysis, SEQ). 

We have amended the title on page 8 to “wellbeing” to provide greater clarity. These item under “wellbeing” are the four ONS Wellbeing questions. ”https://www.ons.gov.uk/peoplepopulationandcommunity/wellbeing/methodologies/surveysusingthe4officefornationalstatisticspersonalwellbeingquestions”

We have scaled these using the standard scaling used by the ONS. This measure is widely used in ONS general population surveys.

6. There is no indication of why a binary measure of race/ethnicity would be valid in this sample. 

It is widely acknowledged that people from BAME groups are more at risk from COVID-19 and thus we wanted to test whether or not a person came from a minority group had any influence on behaviours harmful to health. We have added clarification about this to page 9. 

“As people form BAME backgrounds are more at risk from COVID-19 we wanted to determine whether they were also more at risk from health risk behaviours.”

7. No research supporting the decision to measure relative deprivation by postcode analysis is provided. 

We used this because this is how the UK Government and the Ministry of Housing and Communities measure relative deprivation. (https://assets.publishing.service.gov.uk/government/uploads/system/uploads/attachment_data/file/853811/IoD2019_FAQ_v4.pdf)

8. Capitalize Poisson 

This has been corrected. 

9. 40.3% is not a majority; it is (apparently) a plurality 

We have removed the term “majority”

10. Critically, the specific dates used to define pre- and post-COVID periods are not provided.

We have clarified in the text on page 7 that before the pandemic refers to prior to the pandemic restrictions coming into force. In the results section we give the dates between which the participants completed the questionnaire (July 2020-December 2020) i.e. during the pandemic. 

“In the questionnaire before the pandemic refers to before the pandemic restriction came into force in the UK on the 23rd March.”

Results 

11. Response rate is not provided (please use standardized response rate from AAPOR or another comparable source).

This has now been provided in the flow chart in Figure 1. 

12. Race/ethnicity was proposed as a binary variable in methods but broken out by additional categories in results; this is inconsistent, and the choice of categories is not well-explained. 

We provide fuller demographics for the reader to have a fuller picture of the ethnicity composition of the study however in the model we categorised these into BAME and Non BAME for the reasons described in point 6. 

13. Table 1 is difficult to read given that counts and percentages are placed in a single column; in addition, standard deviations for measures are not provided. 

It is standard reporting to quote ‘n’ and ‘%’ in the same column. Where measures are continuous we have provided the mean and standard deviation. For catergorical variables were quote ‘n’ and ‘%’ for each category. 

14. Table 2 has similar issues, in addition, placing pre- and post-COVID variables vertically rather than horizontally makes it difficult to for the reader to assess changes. 

As the pre and post COVID variables are not the same it was unfortunately not possible to place them horizontally side by side. 

15. Table 4: unclear why univariate models are included at all; table formatting is challenging to read (eg CIs spread across multiple lines, significant associations not distinguished from non-significant with asterisks/bolding, spacing spreads results across multiple pages), and dependent variable is not explicitly stated. 

Thank you for this advice, we have deleted the results of the univariate model and the CIs are now all on one line. In addition we have asterisked the significant results. 

16. Associations between health-harming behaviors and deterioration raised questions of causality. 

We agree with this comment and acknowledge that we cannot determine causality. 

“We do not know whether an increase in health risk behaviours led to a deterioration in physical health or whether a deterioration in physical health led to people being less able to take care of themselves, and this relationship is likely to be bidirectional.”

17. It is unclear to what extent these behaviors would be unique to people with SMI.

We agree that is it unclear to what extent these behaviours would be unique to people with SMI. We make comparisons to the general population throughout the discussion to provide context to the reader. 

Discussion 

18. Comparisons in discussion suggest decreases in physical activity/increase in smoking but do not test for statistical significance, making it unclear whether these are meaningful. The sources of these claims are not linked to the results section (where the tests for significance should be reported). 

We were not able to test the changes in physical activity and smoking for significance as we didn’t have a pre and post measure. We captured the change via a single item for each behaviour which read “please let us know how each of the following habits might have changed since the pandemic restrictions began” with the possible responses “I don’t do that in general”, “More than usual”, “About the same” or “less than usual”. 

19. Links should be included as references not in the text of the manuscript. 

We have included the links as references. 

Conclusions 

20. re: the need for health services are not clearly linked to results.

Thank you we have removed the word ‘health’ and now state that there is a need for services that target health risk behaviours, we believe this to be the case as any increase in health risk behaviour needs to be addressed especially in a group already at risk from higher levels of health risk behaviours and the subsequent effects on both physical and mental health. Health risk behaviours are in part what is leading to the mortality gap people with SMI currently experience. 

Data availability

21. PLOS 1 historically requests that data underlying analysis be made available through a public repository, or, if held by an organization that limits access to data for reasons of confidentiality, for example a government agency that retains protected health information, that a detailed description of how to submit a formal data request and obtain a standardized dataset be provided, e.g. through a website with an explanation of what forms must be completed and what variables to request. The statement that data are “available upon request from the corresponding author” does not meet either standard.

Unfortunately we do not have ethical permission to upload patient level data to a public repository. We will discuss with PLOS ONE what we are able to do. 

Reviewer Two 

Abstract 

1. Some information on the participants features should be provided (sex, age)

We have added an additional sentence to the abstract with this information. 

“The mean age was 50.5 (range = 20 to 86, SD ± 15.69) with 51.0% male and 77.4% white British”

2. There is no significant association between younger and an increase in health-risk behaviours since the IC comprises 1. This is not worth mentioning here.

Thank you, we have removed the reference to age as a significant association from the abstract. 

Introduction 

3. It is mainly well-conducted, but I think that you should mention more relevant studies to support your argument. For instance, since the pandemic has affected health-risk behaviours in general population, this might be exacerbated in people with mental illness. Also, as a consequence of a lack of physical activity, mental health might be deteriorated

https://pubmed.ncbi.nlm.nih.gov/32581985/

https://pubmed.ncbi.nlm.nih.gov/32793013/

Thank you these references have now been included. 

4. An initial hypothesis based on your rationale should be provided 

In this study we sought to explore the health risk behaviours of people with SMI during COVID-19. The study is an exploratory study and as such is not derived from any theoretical framework, therefore we had no specific hypothesis about how health risk behaviours in people with SMI might have changed. We have clarified this at the end of the introduction. 

“This is an exploratory study rather than being derived from a theoretical framework.”

5. Could you provide more information on your sampling framework. It seems as if you have been selecting individuals by those mentioned criteria, but that is not sampling. 

We have created a flow diagram and added some additional information about the sampling to the manuscript. See response to Reviewer One comment 3. 

6. According to the procedure, only those who accepted were included in the study, thus there is a participation rate, I guess.

Yes, however we have now provided additional detail, see response to previous comment. 

7. Were the questions to assess health-related behaviours or self-reported changes validated? 

Questions to assess health related behaviours were self-reported, see response to Reviewer One comment 2.

8. The way you explain why you use Poisson regression is too wordy. You don´t need to explain your prior plans 

We have included this for transparency as we have deviated from the plans in our pre-published analysis plan to prevent any future criticism about deviations from our plan. 

9. Why did you select such covariates? 

The aim of the analysis was to explore the association between the changes in health risk behaviours and changes in physical and mental health adjusting for key sample characteristics. We selected these variables based on experts in the field, statisticians and our lived experience panel.

10. Did you have missing values? 

We have added details to Tables 1 and 2 regarding missing values.

Results 

11. When you report decimals in your Tables, you should stick to always the same number of those.

Thank you for spotting this, this has now been corrected. 

12. Table 4 needs to report the confounders you adjusted your models 

Table 4 previously reported both the univariate and multivariate models and so the associations in the univariate models for each independent variable were adjusted for the presence of the other variables in the model. Following the advice of Reviewer One we removed the univariate models from Table 4. 

Discussion 

13. Your study has three clear bias that should be mentioned: recall bias, selection bias (not real sampling), and causal bias, since your design does not allow to reach causes 

We have added an additional point to the limitations about recall bias (see response to reviewer one, point 2). We used a sampling framework to sample participants from our cohort, rather than use a convenience sample, we have acknowledged in the limitations section that we may not have been able to contact the most vulnerable section of the SMI population and the efforts we went to, to do all we could to include them. We have not sought to make any claims about causation and have clarified this in the limitations section. 

“Thirdly, the design used in this study does not allow us to determine the causes of any changes in health related behaviours so we cannot be certain as to why behaviours might have changed.” 

Conclusion 

14. It is the first time I see citations in a Conclusion 

We wanted to include these citations as we felt it was important for the reader to realise that we were not making statements that were not backed up by evidence.

---

## [Decision Letter · Decision Letter 1]

27 Sep 2021

Health risk behaviours among people with severe mental ill health during the COVID-19 pandemic: analysis of linked cohort data

PONE-D-21-09538R1

Dear Dr. Peckham,

We’re pleased to inform you that your manuscript has been judged scientifically suitable for publication and will be formally accepted for publication once it meets all outstanding technical requirements.

Kind regards,

Jagdish Khubchandani

Academic Editor

PLOS ONE

Additional Editor Comments (optional):

Thank you for the revisions and clarifications.

Reviewers' comments:

Reviewer's Responses to Questions

**Comments to the Author**

1. If the authors have adequately addressed your comments raised in a previous round of review and you feel that this manuscript is now acceptable for publication, you may indicate that here to bypass the “Comments to the Author” section, enter your conflict of interest statement in the “Confidential to Editor” section, and submit your "Accept" recommendation.

Reviewer #2: All comments have been addressed

2. Is the manuscript technically sound, and do the data support the conclusions?

Reviewer #2: Yes

3. Has the statistical analysis been performed appropriately and rigorously? 

Reviewer #2: Yes

4. Have the authors made all data underlying the findings in their manuscript fully available?

Reviewer #2: Yes

5. Is the manuscript presented in an intelligible fashion and written in standard English?

Reviewer #2: Yes

6. Review Comments to the Author

Reviewer #2: (No Response)

7. PLOS authors have the option to publish the peer review history of their article (what does this mean?). If published, this will include your full peer review and any attached files.

Reviewer #2: No

---

## [Editor Report · Acceptance letter]

5 Oct 2021

PONE-D-21-09538R1 

Health risk behaviours among people with severe mental ill health during the COVID-19 pandemic: analysis of linked cohort data 

Dear Dr. Peckham:

I'm pleased to inform you that your manuscript has been deemed suitable for publication in PLOS ONE. Congratulations! Your manuscript is now with our production department. 

Kind regards, 

on behalf of

Dr. Jagdish Khubchandani 

Academic Editor

PLOS ONE